# Variation in Root and Shoot Growth in Response to Reduced Nitrogen

**DOI:** 10.3390/plants9020144

**Published:** 2020-01-23

**Authors:** Seth Tolley, Mohsen Mohammadi

**Affiliations:** Agronomy Department, Purdue University, 915 West State Street, West Lafayette, IN 47907, USA; tolleys@purdue.edu

**Keywords:** wheat, root at seedling stage, root at mature stage, root growth pattern, nitrogen uptake

## Abstract

Recently, root traits have been suggested to play an important role in developing greater nitrogen uptake and grain yield. However, relatively few breeding programs utilize these root traits. Over a series of experiments at different growth stages with destructive plant biomass measurements, we analyzed above-ground and below-ground traits in seven geographically diverse lines of wheat. Root and shoot biomass allocation in 14-day-old seedlings were analyzed using paper roll-supported hydroponic culture in two Hoagland solutions containing 0.5 (low) and 4 (high) mM of nitrogen (N). For biomass analysis of plants at maturity, plants were grown in 7.5 L pots filled with soil mix under two nitrogen treatments. Traits were measured as plants reached maturity. High correlations were observed among duration of vegetative growth, tiller number, shoot dry matter, and root dry matter. Functionality of large roots in nitrogen uptake was dependent on the availability of N. Under high N, lines with larger roots had a greater yield response to the increase in N input. Under low N, yields were independent of root size and dry matter, meaning that there was not a negative tradeoff to the allocation of more resources to roots, though small rooted lines were more competitive with regards to grain yield and grain N concentration in the low-N treatment. In the high-N treatment, the large-rooted lines were correlated to an increase in grain N concentration (r = 0.54) and grain yield (r = 0.43). In low N, the correlation between root dry matter to yield (r = 0.20) and grain N concentration (r = −0.38) decreased. A 15-fold change was observed between lines for root dry matter; however, only a ~5-fold change was observed in shoot dry matter. Additionally, root dry matter measured at the seedling stage did not correlate to the corresponding trait at maturity. As such, in a third assay, below-ground and above-ground traits were measured at key growth stages including the four-leaf stage, stem elongation, heading, post-anthesis, and maturity. We found that root growth appears to be stagnant from stem elongation to maturity.

## 1. Introduction

During the Green Revolution, from 1960 to 1985, global wheat yields increased from 1088 to 2172 kg/ha (~100% increase); however, from 1985 to 2010, global wheat yields only increased from 2172 to 2972 kg/ha (~37% increase) [1]. From 1985 to 2010, only 62% of wheat acres experienced yield increases [2]. Approximately 38% of global wheat acreage experienced no yield growth, while 1% of the acreage experienced decreased yields. To meet future demands, wheat production must be doubled by 2050 [3] without an increase in area harvested through deforestation [4], harvest frequency [5], and water and nutrient demands [6]. That requires increased yields of 2.4% per year [7], while from 1985 to 2017, yields only increased by 1.6% per year [1].

From 1961 to 2007, increased wheat yields were correlated to increased N inputs (R^2^ = 0.89) [8]. While yields have increased, N loss has become a prevalent topic in the agriculture industry [9,10,11]. In high-N environments, Raun and Johnson [11] found that only 33% of the applied N is taken up and remobilized into the grain of cereal crops. The remaining N is lost through denitrification, volatilization, and leaching or left in biomass at harvest [9]. In wheat, an increased N rate from 150 to 250 kg N ha^−1^ led to an average of 36% of the extra N left in the soil and susceptible to greater amounts of N leaching from the root zone [10]. When further increased from 250 to 350 kg N ha^−1^, almost 90% of the additional N was left in the soil and at risk of leaching.

On the other hand, many wheat-growing regions with fewer resources that cannot apply fertilizer result in reduced leaf area index, above-ground biomass, tiller number, kernel number, and ultimately grain yield [12]. Kharel et al. [13] found a decrease in yields of 28% in environments with 0 N fertilization compared to 160 kg N ha^−1^.

To increase yields in these environments, it was suggested by Lynch [14] that a greater understanding and utilization of below-ground, root traits are key for a ‘second Green Revolution’. Previous experiments identified important root traits in both high- and low-N environments. In high-N environments, increasing root biomass [15] and root length density [16,17] were shown to be associated with greater N uptake and yield. On the other hand, studies in N-deficient environments report that increased ‘early vigor’ [18], the root:shoot ratio [19], and lower specific root length (root dry matter per unit length) [20] are associated with greater productivity. To identify these traits, we used a diverse set of lines to obtain preliminary conclusions about potentials of these phenotypes in the determination of grain yield. The objective of this functional root phenotyping study was to examine the genetic variation in root dry matter under different N treatments and assess the relationship between structural and functional impact of root dry matter on yield and nitrogen uptake.

## 2. Materials and Methods

### 2.1. Plant Materials

Seven experimental breeding lines and landraces of wheat (*Triticum aestivum* L.), with spring growth habit, were used in this study. The lines were from various geographical origin or breeding geographical locations, namely, Opata, originally from Mexico; PI 189823, from, Santa Fe, Argentina; PI 245427, from Afghanistan; PI 519677, from Chile; PI 542457, from Nebraska, United States; PI 626655, from Iran; PI 94379, from Armenia. Seven lines were studied at seedling stage and maturity stage assays and four of them were studied for root analyses at further growth stages.

### 2.2. Seedling Assay

In a growth chamber and paper roll-supported hydroponic experiment, the biomass allocation patterns of these lines were examined under high- and low-N treatments at the seedling stage. Six uniform seeds from each line were grown in kraft paper (30 cm wide and 45 cm long) utilizing an experimental procedure explained by Rahnama et al. [21]. Before the seeds were ‘planted’, the paper was rolled by making a 1.5 cm crease for seed placement and a 3 cm crease along the side for further rolling. Seeds were treated with Maxim XL fungicide (100 μL/L) (active ingredients: fludioxonil and mefenoxam) and evenly placed in the kraft paper with the embryo facing the bottom of the paper. To ensure the seeds would remain in place, the paper was rolled tightly. Since the kraft paper was not good at soaking up water, each roll was covered with a moist paper towel. Individually, the rolls were placed into a PVC tube (5.2 cm wide by 38.5 cm long) and filled with water or later with nutrient solution. Representative hydroponic tubes containing the 7 lines are presented in (Figure 1A). To reduce evaporation, the tubes were wrapped with parafilm. The growth chamber was set to 20 °C and a 12 h photoperiod with a relative humidity of 50% for the duration of the assay. The photosynthetic photon flux reached 300 μmol m^−2^ s^−1^ at the top of the plant canopy. Upon the coleoptiles appearing from the tips of the paper in all lines, five days after planting, the plants were exposed to a modified Hoagland solution. This solution contained: KH_2_PO_4_, 0.2 mM; MgSO_4_, 0.5 mM; CaCl_2_, 0.5 mM; H_3_BO_3_, 1 × 10^−3^ mM; (NH_4_)_6_MO_7_O_24_, 5 × 10^−5^ mM; CuSO_4_, 5 × 10^−4^ mM; ZnSO_4_, 1 × 10^−3^ mM; MnSO_4_, 1 × 10^−3^ mM; Fe(III)–EDTA, 0.1 mM. The pH of the solution was adjusted using HCl and KOH to 6.2–6.3. The varying N levels, (0.5 mM NO_3_^−^ and 4.0 mM NO_3_^−^), were established by adding Ca(NO_3_)_2_4H_2_O and will be called low-N and high-N from this point forward [22,23]. To maintain the treatments, nutrient solutions were refreshed every two days.

Shoot and root samples were collected 14 days after planting (DAP), and roots were analyzed by WinRhizo software (Figure 1B,C). Roots were preserved in 50% ethanol. Roots and shoots were oven dried at 60 °C for two days and shoot dry matter and root dry matter were recorded. Shoot N concentration was measured in shoot dry matter to verify the different N treatments. Nitrogen concentration was assessed through dynamic flash combustion using a Thermo Scientific FlashEA 1112 Elemental Analyzer (CE Elantech, Lakewood, NJ). Briefly, in dynamic flash combustion, samples are heated to 900–1000 °C, and the organic and inorganic material is converted into elemental gases once oxygen is added [24].

This observation nursery was conducted in two side-by-side experiments. The six reps of a given line were in the same PVC tube, and therefore, this experiment did not allow for analysis of variance (ANOVA) to test for differences between lines, N treatments, or the line × N interaction. Instead, to evaluate the difference between N treatments, permutation tests were performed between the means of the N treatments by resampling 10,000 times.

### 2.3. Maturity Assay

A follow-up experiment was conducted on the same 7 lines in the greenhouse to evaluate below-ground and above-ground traits and their relationship under high- and low-N treatments. Seeds were germinated in petri dishes and transplanted (one seedling per pot) into 7.5 L pots containing a 1:1:1 mix of topsoil (Biotown Ag, Reynolds, IN), sand, and potting soil (Sungro Metro-mix 510). Plants were bottom-watered with tap water three times a week. The temperature was set to 23 °C and the photoperiod was 12 h for the duration of the experiment. Starting one week after transplanting, the same low-N and high-N solutions were applied weekly at a rate of 125 mL week^−1^ for 35 DAP and 250 mL week^−1^ from 42 DAP to maturity.

At maturity, the below-ground and above-ground traits were measured destructively. Above-ground traits include duration of vegetative growth, plant height, shoot dry matter, tiller number, kernel number, yield, and grain N concentration. Duration of vegetative growth was measured as the number of days from planting to heading. A ruler was used to measure plant height, while tiller number was counted by hand. Spikes were harvested and the above-ground tissue was cut at the crown level. Spikes and above-ground tissue were dried for two days at 60 °C and dry matter was recorded. Spikes were then hand-threshed, and kernel number and yield were measured for each plant. For below-ground traits, roots where removed from the soil after the above-ground traits were collected. Roots (and soil) were submerged in a tub of tap water for ~30 min, then washed and rinsed with a medium-pressure hose several times. The clean roots were placed in 50% ethanol for future analysis, and later were dried at 60 °C for two days and root dry matter was recorded. For measuring root dry matter, roots were cut off crown ~1–2 cm away from crown (Figure 2). The ratio of root-shoot was measured in each replicate by dividing shoot dry matter by root dry matter whereas rooting efficiency (measured on the basis of phytomass production per unit root dry matter or on the basis of grain yield production per unit root dry matter) were derived by dividing shoot dry matter or grain yield, respectively, by root dry matter. Grain nitrogen concentration was assessed for a composite for each line in each environment with 2 replicates to assess the difference in the N treatments.

This assay was conducted in a split-plot design with eight replicates. N treatment was the whole-plot effect in two levels and line as the sub-plot in seven levels. ANOVA using type 3 SS was performed for N effect, line effect, and the line × N treatment effect. The N effect was tested against the main-plot error (N × Rep interaction) while line and the line × N interaction were tested against the sub-plot error (residual MSE). For traits with a significant Line × N interaction, Tukey tests were performed on the interaction. For traits where the Line × N interaction was not significant, but line was significant, Tukey tests were performed to differentiate lines. Tukey tests were performed using agricolae: HSD test. Repeatability was by looking at the amount of variation explained by line compared to the total amount of variation for each trait.

### 2.4. Temporal Assay

Two small root dry matter lines (PI 189823 and PI 519677) and two large root dry matter lines (PI 542457 and PI 626655), based on the results of maturity assay, were selected for the growth stage experiment. The goal was to further evaluate below-ground and above-ground traits and temporal patterns of biomass allocation. Single plant per 7.5 L pots were grown in a 1:1:1 mix of topsoil (Biotown Ag Reynolds, IN), sand, and potting soil (Sungro: propagation mix). The temperature was set to 23 °C and the plants received a 12 h photoperiod for the duration of the experiment. Nutrient solution (high N) was the same for all plants in this assay.

Destructive sampling was performed at five growth stages: four-leaf stage, stem elongation, heading, post-anthesis, and maturity. Sampling began as each plant reached the desired growth stage and was recorded as DAP. For each plant, phenotypes measured include plant height, shoot dry matter, and root dry matter, similar to the maturity assay. For destructive measurements at heading, post-anthesis, and maturity, emerged spikes were cut away from the above-ground biomass after plant height was measured.

The growth rate analysis experiment was completed in a randomized complete block design (RCBD) with four replicates at the four-leaf stage and five replicates in all other stages. Lines, growth stage, and their interaction were considered as fixed effects while replicates was considered as random effect. ANOVA using type 3 SS was performed to assess the significance of the fixed effects. For traits with a significant line x growth stage interaction, Tukey tests were performed on the line × growth stage interaction using lsmeans: lsmeans and multcomp: cld. However, when the interaction of line × growth stage was not significant, significant differences between the levels of each line and growth stage were further investigated using Tukey tests. All analysis was performed in R environment [25].

## 3. Results

### 3.1. Considerable Decrease in Shoot Dry Matter but Variable Root Responses to Low-N Treatment in the Seedling Stage

This study aimed to examine the genetic variation in root dry matter under contrasting N treatments in seedling and maturity stages and associate root dry matter to N uptake and utilization, above-ground biomass and grain traits. To characterize seedling traits, paper roll-supported hydroponic system and WinRhizo system were used (Figure 1). Shoot and root dry matters were in the range 17.6–53.1 mg and 10.4–20.6 mg, respectively, in high-N treatment. In response to low nitrogen treatment, seedlings’ shoot dry matter decreased in all lines by an average of 70%. Through a permutation test, this 70% decrease was found to be beyond the levels (*p*-value < 0.01) that could be caused at random. In contrast to invariable decreases in shoots, the changes of the root dry matter and length to low N was not consistent. The average root dry matter in high N was 15.3 mg (range 10.4–20.6 mg) and the average root dry matter in low N was 16.3 mg (range 11.6–21.1 mg). Five lines showed increases in root length and dry matter while two lines showed decreases in these phenotypes (Figure 3). As a result of considerable decreases in shoot dry matter, the root-shoot ratio increased in all lines in response to low-N treatment. We also observed differential nitrogen concentration in the seedlings. The shoot nitrogen concentration in high N was 0.043 g of nitrogen per gram of tissue dry matter (g N/g DM) while in low N it was 0.027 g N/g DM. Through a permutation test, this decrease was found to be beyond the level (*p*-value < 0.001) that could be caused at random.

### 3.2. Substantial Correlation of Root with Key Above-Ground Traits at Maturity Stage

Analysis of variance revealed that the effect of N treatment was significant only for grain N concentration but not for any other traits (Table 1). A possible explanation for these results is that in the maturity test, the pots were filled with a potting mix including organic materials (Sungro Metro-mix 510) and top soil from Indiana, which might have enough N to support early stages of plant growth in our experiment. Therefore, it is possible that during early stages plants were not really subjected to differential N treatment. It was probably during maturity and grain production that the existing N in the soil is depleted and a relatively differential N treatment is established by nutrient solutions. Unfortunately, we did not measure N level in the potting medium to present in this report. Lines were significantly different for duration of vegetative growth (days to heading), plant height, shoot dry matter, root-shoot ratio, root dry matter, root efficiency for phytomass, root efficiency for grain yield, and kernel number. The interaction of Line x N was significant only for plant height and shoot dry matter but no other traits. The significance levels and heritability of traits analyzed in the maturity assay are presented in Table 1. In high-N treatment, duration of vegetative growth ranged from 38 to 56 DAP. In low N, the duration of vegetative growth, on average, was 3 days longer although not significant. Shoot dry matter averaged 3.07 g in high N with a range of 0.98 to 5.49 g (sd of 1.83). Shoot dry matter decreased by ~10% in the low N to an average of 2.77 g, with a range of 1.03 to 4.34 g (sd of 1.30). The decrease in shoot dry matter was more pronounced in genotypes with greater dry matter in high N, though the overall effect of the N treatment was not significant due to the variability in the line response to the N treatment and the standard deviation within each N treatment. (Figure 3).

Three traits showed strong and significant correlations with shoot dry matter. These were the duration of vegetative growth, tiller number, and plant height (Figure 4). The correlation between duration of vegetative growth and shoot dry matter was r = 0.57 in high N and r = 0.53 in low N. Tiller number was correlated (r = 0.83 and 0.68) with shoot dry matter in high and low N, respectively. The correlation between plant height and shoot dry matter was r = 0.72 in high N and 0.75 in low N.

Root dry matter ranged from 0.12 to 1.81 g, with an average of 0.78 g in high N, and showed variable responses among various lines to low N (Figure 3 and Figure 5). Similar to the dependency of shoot dry matter, we observed considerable correlations between root dry matter and the key phenological and anatomical traits. The correlation of duration of vegetative growth, tiller numbers, and plant height with root dry matter were 0.81, 0.82, and 0.50 in high N (Figure 4). The same numbers were 0.79, 0.70, and 0.54 in low N. Shoot dry matter, itself, was correlated to root dry matter r = 0.84 in high N and 0.80 in low N (Figure 4).

### 3.3. Distribution of Dry Matter between Shoot and Root at Maturity

In the maturity assay, root dry matter contributed to on average ~20% of total plant biomass in both N treatments. The root-shoot ratio range was 8%–33%, averaging 21% in high N. The contribution of root dry matter to the entire biomass range was 3%–35%, averaging 20% in low N. The effect of N treatment was not significant on the root-shoot ratio (*p* value > 0.05). However, lines were significantly different (*p* value < 0.001) in their root-shoot ratio.

At maturity, the genetic variation below ground was greater than the amount of variation above ground. Root dry matter in the high N and low N was 0.78 and 0.68 g, respectively. Significant differences (*p* value < 0.001) for root dry matter were found among the lines with a range of 0.12–1.81 g in high N and 0.09–1.56 g in low N. This difference corresponded to a ~15-fold difference in root dry matter, which was far greater than the ~5-fold difference in shoot dry matter. In addition, root dry matter was correlated to the root-shoot ratio with r = 0.87 in high N and r = 0.92 in low N, which indicates that plants are able to alter their below-ground strategies and phenotypes more freely than the above-ground phenotypes.

There was evidence that the lines had various biomass allocation strategies in response to the change in the N treatment. In Figure 5, there are four groups of lines that were judged in this experiment. PI 542457 (5), PI 626655 (6), and PI 94379 (7) increased shoot and root dry matter in response to the added N input which also led to an increase in the grain yield that was observed for these lines. PI 189823 (2) and PI 519677 (4) had relatively consistent root and shoot dry matter across the N treatments which led to minimal differences in grain yield across the N treatments. Opata, which maintained a consistent biomass allocation between root and shoot dry matter, showed an increase in grain yield when grown in low N. Finally, PI 245427 increased the amount of shoot and root dry matter in the low-N treatment, which led to an increase in the grain yield in the low-N treatment. However, yield in this experiment was based on the grain produced from a single plant rather than field trials and with plant-to-plant interaction. In Figure 5, there is a linear pattern in the responses of shoot dry matter as a function of changes in root dry matter, indicating an equilibrium that can likely be explained by the high correlation of root dry matter and shoot dry matter in both the high-N and low-N treatment.

Our data indicated that the functionality of large roots in nitrogen uptake was dependent on the availability of N. In the high-N treatment, the large-rooted lines were correlated with a greater grain N concentration (r = 0.54) and grain yield (r = 0.43) (Figure 5). In other words, in high N, the lines with the greatest root dry matter (PI 542457, PI 626655, and PI 94379) also had the three greatest yields and three of the four greatest grain N concentrations. In low N, the correlation of root dry matter to yield and grain N concentration both decreased (r = 0.2 and r = −0.38, respectively). In low N, the highest yields came from PI 626655, PI 245427, Opata, and PI 189823 (Table 1). PI 626655 had the most root dry matter (2.69 g), PI 245427 had an intermediate amount of root dry matter (0.91 g), and Opata and PI 189823 had small amounts of root dry matter (0.15 and 0.09 g, respectively). This variation in N responses under low N environment suggests that yield is not correlated with root dry matter in low N and therefore, there is potential to screen wider genebank germplasm for identifying ideal root-shoot relationship. Additionally, we found that two of the three lines with the greatest grain N concentration were Opata and PI 189823, which had some of the smallest root dry matter in low N.

### 3.4. Temporal Assay and Root Growth Rate Analysis

The correlation of root dry matter between seedling and maturity stages was r = −0.66. Since this correlation was negative, we aimed to further characterize root growth patterns during progressive growth stages. In this assay, the line x growth stage interaction was significant for root dry matter, which indicated that lines had different patterns of biomass allocation. At the four-leaf stage, there was not a significant difference among lines for root dry matter; however, by stem elongation, root dry matter of PI 626655 was significantly greater than the other three lines. PI 519677 had not significantly increased root dry matter at stem elongation over that observed at the four-leaf stage. Invariably across all four lines, root dry matter did not significantly change from stem elongation to maturity (Figure 6). A pronounced observation of this behavior is that root dry matter at stem elongation explained 99% of variation of root dry matter at maturity, indicating that root growth appeared to stagnate after stem elongation regardless of the root system size.

## 4. Discussion

### 4.1. Structural Variation of Root Traits

Measuring roots under natural conditions by digging out plants is the gold standard for root phenotyping [26]. However, evaluating root phenotypes in less heterogeneous environments, including those of paper roll-supported hydroponic [27,28] and sand-culture hydroponic [29] systems, that allow single-plant analysis may provide plant roots with the opportunity to fully express their genetic potential irrespective of nutrient availability. Highly correlative traits can be measured interchangeably to save time and effort. A previous experiment has shown a correlation of r = 0.8 between root dry matter and root length in wheat seedlings of 200 historical wheat accessions [30], which indicates that root dry matter can be used as indirect estimation of total root length. Root architecture refers to the spatial configuration of the root system within the soil and is characterized by vertical and horizontal extending and proliferation, usually by three-dimensional presentation and quantification of traits such as length and angles. We extracted roots from the soil by washing. Representative root images of lines under differential N treatments are shown in Figure 7. The lack of structural stability in the root systems, after extracting them from the soil by washing (Figure 7), did not allow for documentation of root angles and orientation. We could, however, demonstrate positioning and orientation of the basal segment of the roots (because they seem to be more stable) (Figure 2).

While there was genetic variation present in the seedling roots, it was smaller to the range observed at maturity, suggesting that the genetic potential for root traits was realized later in the development. Narayanan et al. [31] evaluated a panel of 297 diverse spring wheat lines in PVC tubes with 7.5 cm diameter and 150 cm height filled with Turface MVP (PROFILE Products LLC, Buffalo Grove, IL). They found a ~35-fold difference in root dry matter from 0.22 to 7.6 g. These results appear to verify the wide genetic diversity of root dry matter in spring wheat. Similarly, in two-week seedlings, Beyer et al. [30] reported a range of 6–20 mg for root dry matter in 215 historical lines evaluated in a paper roll-supported hydroponic system.

One of the goals of these projects was to evaluate biomass allocation patterns under varying N inputs. While N treatment was only significant for grain nitrogen concentration, we believe that the large variability between lines, which lead to a high standard deviation within N treatments, was a main contributor in the insignificance of N treatment for many of these traits. Nevertheless, among these seven lines, there appeared to be four biomass allocation patterns (Figure 5). PI 542457, PI 626655, PI 94379 increased shoot and root dry matter in high N which led to the increase in grain yield. PI 189823 and PI 519677 retained consistent biomass allocation and yield across N treatments. Opata had consistent biomass allocation across N treatments though yield increased in low N. PI 245427 increased both root and shoot, which resulted in an increase in grain yield in low N. Among these seven lines, there appears to be a fairly linear trend in Figure 5, where the response to low N was evident in both shoot and root dry matter. As such, we do not observe biomass allocation patterns in the top, left or bottom, right quadrants of Figure 5. It would be interesting and beneficial to find lines that deviate from this linear pattern to see the associated change in yield across N treatments.

### 4.2. N Recovery and Utilization as a Functional Trait

Despite studies concerning structural root traits, the amount of experimental evidence that demonstrates a link between structure and function is limited. In this context, root efficiency can be expressed as a variety of measures. For example, phytomass production per unit root dry matter and uptake of nitrogen per unit root length. A question before breeding specialists is whether crop plants can be improved for functional and efficiency traits for example selecting for greater efficiency of uptake and transport to the shoot. A proof of concept study in rice has demonstrated up to 3-fold variation in total biomass at a given root size, highlighting that genotypes are, indeed, different in their efficiency in uptake of nutrient - in this case phosphorous [32]. In the current study, at seedling stage the shoot dry matter per unit root dry matter ranged from 0.7 to 3.73 g/g in high N and 0.7–3.27 g/g in low-N treatment. At maturity, our data revealed that yields were independent of root size and dry matter under low-N treatments, meaning that there was not a negative tradeoff to the allocation of more resources to roots. Under high N, plants with larger roots had a greater response to the increase in N input. In high N, there was a trend that the large-rooted lines (PI 542457, PI 626655, and PI 94379) also had high yields and grain N concentration. In low N, there were great variation between root dry matter and yield. Nevertheless, it was observed that Opata and PI 189823 which had 2 of the 3 smallest root dry matter were among the top performing lines for yield and grain N concentration. In is interesting to note that these are two of the lines with relatively unchanged biomass allocation between the N treatments. Though, in low N, the line with the highest yield and among the highest grain N concentrations was PI 626655, which lead to the inconsistent trend.

This result was also seen in a high-N treatment by Ehdaie and Waines [15], who observed that large-root systems had the potential to increase both yield and grain protein content while reducing N pollution. However, our results conflict with those by Lynch [14] in which large root dry matter was counterproductive in low-N treatments as the metabolic cost of maintaining the large root was a resource drain. It must be noted that in the current study, the correlation of root dry matter and tiller number was substantial and significant. Therefore, an ideal situation for declaring a significant relationship between root dry matter and grain yield would be comparisons of grain yields between two lines with nearly equal root dry matter.

### 4.3. Implications of Trait Correlations for Future Genetic Research

Phenotypic correlation among traits arise from shared genetic and environmental effects. The genetic association among traits cast doubt on the interpretation of genetic studies. Conditional genetic mapping was suggested as one solution for the inheritance analysis of correlated traits [33]. Our study revealed tight correlations among duration of vegetative growth, tiller number, shoot dry matter, and root dry matter. For example, in the maturity stage, the correlation between root dry matter and tiller number was 0.82 and 0.7 in high N and low N, respectively (Figure 4). When two contrasting individuals for root traits are chosen for developing bi-parental mapping population, it is likely that these two lines are simultaneously contrasting for tiller numbers. Therefore, it is expected that these two traits are co-inherited to progeny together. Any attempt for genetic mapping of root traits may result in the identification of genomic regions that also control tiller numbers. Conditional trait mapping is a suitable approach in these cases. For example, Zhang et al. [34] performed associations on a primary trait (kernel weight) while conditioning on a secondary trait (spike number).

A potentially interesting and long-standing question is the effect of dwarfing genes on root traits. During the Green Revolution, breeders developed semi-dwarf, high-yielding wheat varieties resistant to lodging by selecting short plants with stiff stalks. Waines and Ehdaie [15] argue that selecting for favorable above-ground traits during breeding may have negatively impacted root traits during the Green Revolution. Several studies investigating the relationship between root traits and reduced height (Rht) genes produced contradictory results. Some groups observed that Rht genes had a slight negative effect on wheat root traits [35,36], while other groups reported the effects were considerable [37,38,39]. Subira et al. [39] observed that the Rht-B1b allele decreased root biomass (per plant) by 28%. Our study, while not presenting allelic variation of Rht genes, showed that a suit of traits including plant height are in tight correlation with root traits.

### 4.4. Correlation of Seedling and Maturity Root Phenotypes and Time-Dependent Genotypic Differences

Growth cycle of wheat follows a continuous progression beginning at germination, proceeding through vegetative growth, flowering, grain fill, and terminating at maturity. For traits collected at different time points such as root at seedling stage and root at maturity stage, genotypes may display higher values of a time-dependent trait in one stage, but lower values in another stages. Moreover, it may be the case that a minimum time threshold is needed for a trait to be expressed to its full potential. Our study showed that small-rooted lines in seedling stage where those that produced the largest root dry matter in maturity stage. For example, root dry matter measured after 2 weeks was negatively correlated (r = −0.66) with those measured at maturity. The temporal assay showed that root dry matter did not significantly increase or decrease from stem elongation to maturity and variation of root dry matter at stem elongation explained nearly 99% of variation at maturity stage (Figure 6).

While this result was based on only four lines, it could cautiously be said that selecting root dry matter at stem elongation was as good as selection at maturity. These results agree with Liu et al. [40]. In a winter-wheat field experiment to evaluate the impact of N and water on root growth at multiple growth stages, they found that 71% of the variation in root weight density at maturity was explained in the variation in root weight density at jointing. Furthermore, they found that 82% of the variation in root weight density at maturity was explained in the root weight density at flowering. In corn, most rapid development of corn (*Zea mays* L.) roots occurs during the first 8 weeks after planting [41]. After silking, corn root length declines [42]. This decline in root length after silking presumably is due to the high carbon (C) demand of grain resulting in enhanced translocation of C and N to grain, including some C and N that roots would normally obtain [43]. Slaton et al. [44] studied root growth dynamics of lowland rice and found that maximum root growth rates were reached between active tillering and panicle initiation, and maximum root length was reached by early booting.

## 5. Conclusions

In conclusion, these assays were performed to evaluate the performance and root characteristics of a few diverse lines. A collection of lines from around the world was phenotyped and a ~15-fold difference was noted between lines for root dry matter, whereas only a ~5-fold difference was measured between shoot dry matter. This diversity is an indication of the great potential that can be achieved if root traits are incorporated into breeding programs. In high N, we found a trend that the top yield and grain N concentration came from lines with the largest root dry matter. In low N, the trend between root dry matter to yield and grain N concentration showed variation. It should be noted though that smaller-rooted lines were more competitive in the low-N treatment than in high N. Furthermore, root dry matter was identified across growth stages (i.e., seedling and maturity) from which selection could be beneficial as it was associated to greater N uptake and yield. Finally, varied biomass accumulation patterns above and below ground were described on a single-plant basis, from which it appears that root dry matter remains consistent from stem elongation to maturity. While above-ground traits have commonly been used for selection in the past, incorporating root traits into breeding programs is an area from which there appears to be great, untapped potential. This potential could help to bridge the gap between current wheat production and the demands of the future, where it is likely that nitrogen management practices will be adjusted to prevent leaching and environmental damage.

## Figures and Tables

**Figure 1 plants-09-00144-f001:**
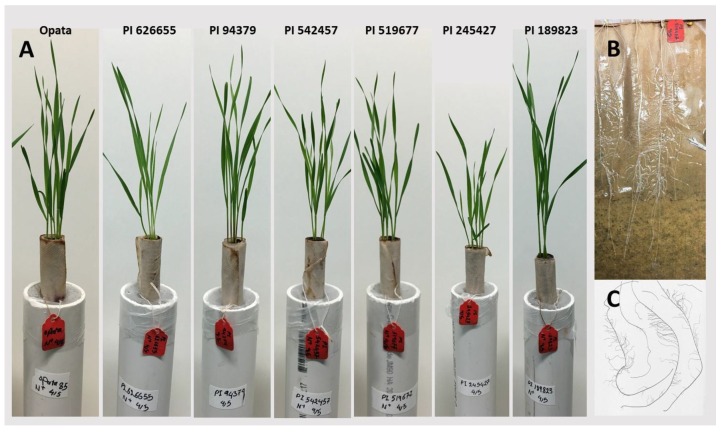
Seedlings were grown for two weeks in paper roll-supported hydroponic solutions containing low or high N nutrient solutions. Representative hydroponic tubes of each line (**A**), unfolded paper roll after two weeks that shows seminal root growth and branching (**B**), and the image taken on tray by the WinRhizo software (**C**) are demonstrated.

**Figure 2 plants-09-00144-f002:**
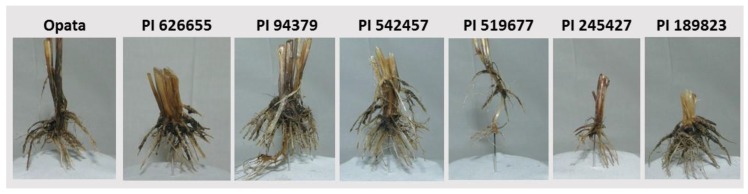
Representative images of crowns after cutting the roots from the 7 lines at maturity.

**Figure 3 plants-09-00144-f003:**
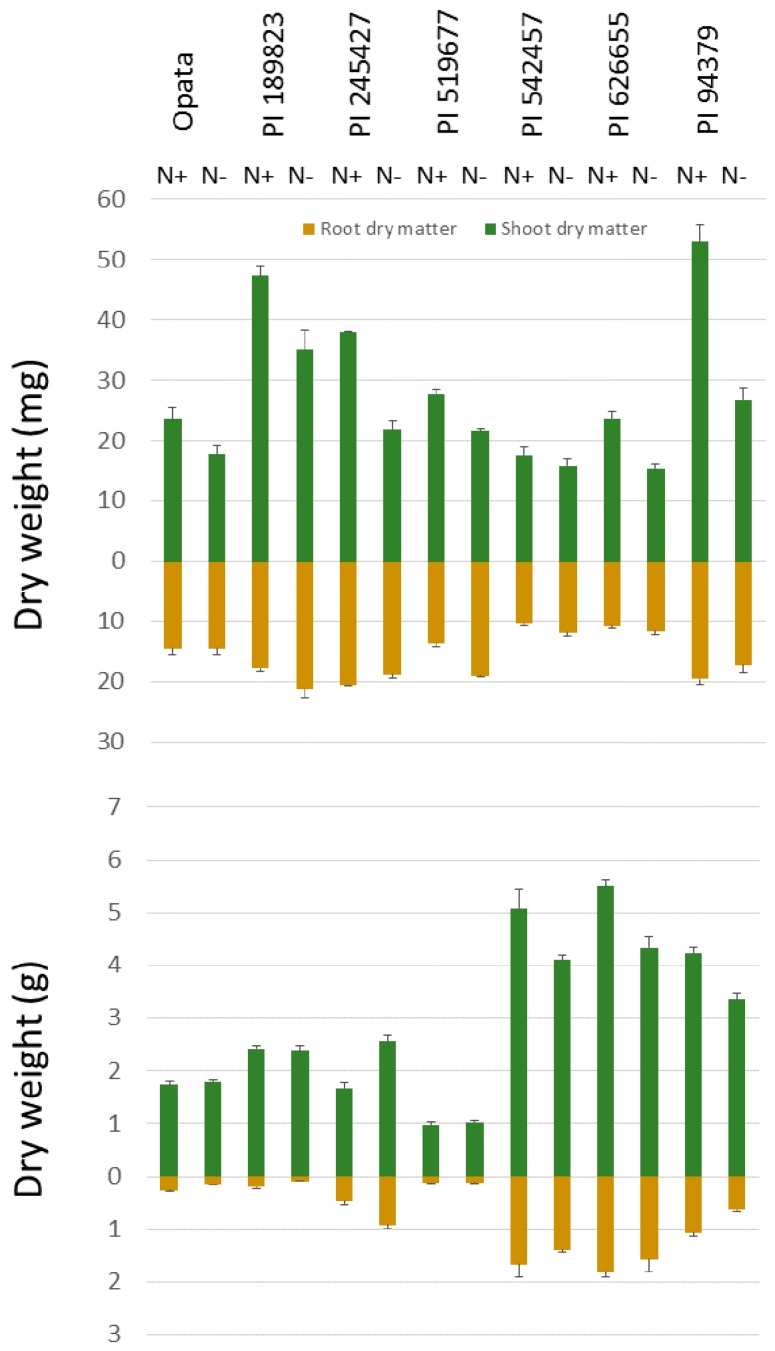
Shoot and root dry matter presented for the seedling (**top**) and maturity (**bottom**) assays. Error bars represent the standard errors of the means for each line in each treatment. The root dry weight is presented below the horizontal axis to demonstrate that the mass recorded is from belowground.

**Figure 4 plants-09-00144-f004:**
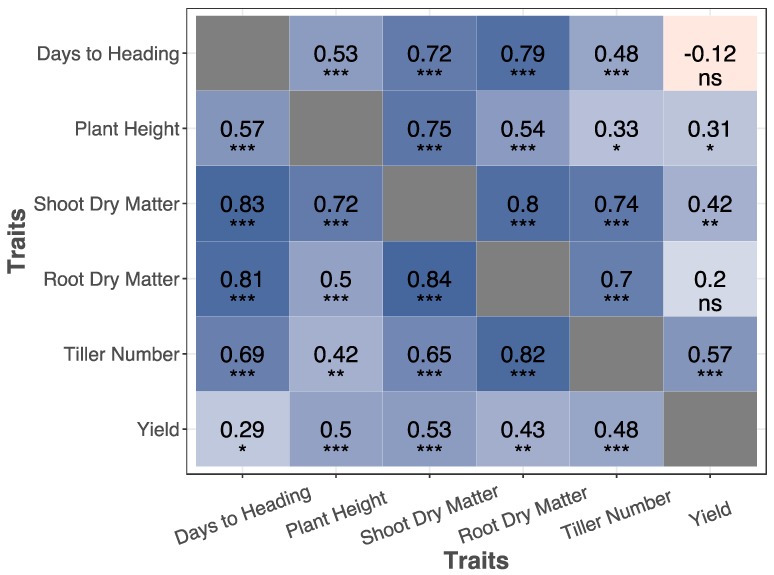
Matrix displaying the correlations and significance of traits measured in the maturity assay. Red heatmap signify a negative correlation between the traits whereas blue heatmap represents a positive correlation. Above the diagonal represents the correlation of these traits in low N whereas below the diagonal represents the correlation of the traits in high N. *p* value significance of as follows: *** < 0.001; 0.001 < ** < 0.001; 0.01 < * < 0.05; ns > 0.1.

**Figure 5 plants-09-00144-f005:**
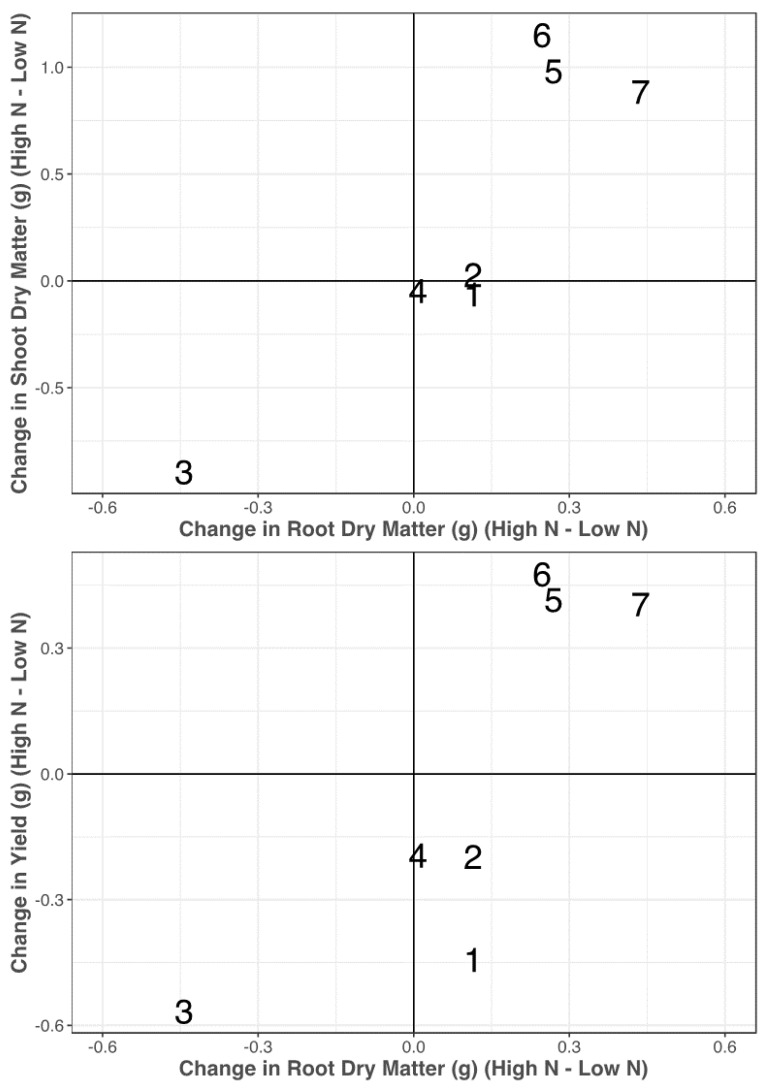
Variation of biomass allocation responses with regard to N treatments (high N–low N). On the x axis is the change in root dry matter with change in shoot dry matter (**top**) and change in single-plant yield (**bottom**) on the y axis. Lines are distinguished as follows: Opata (1), PI 189823 (2), PI 245427 (3), PI 519677 (4), PI 542457 (5), PI 626655 (6), and PI 94379 (7).

**Figure 6 plants-09-00144-f006:**
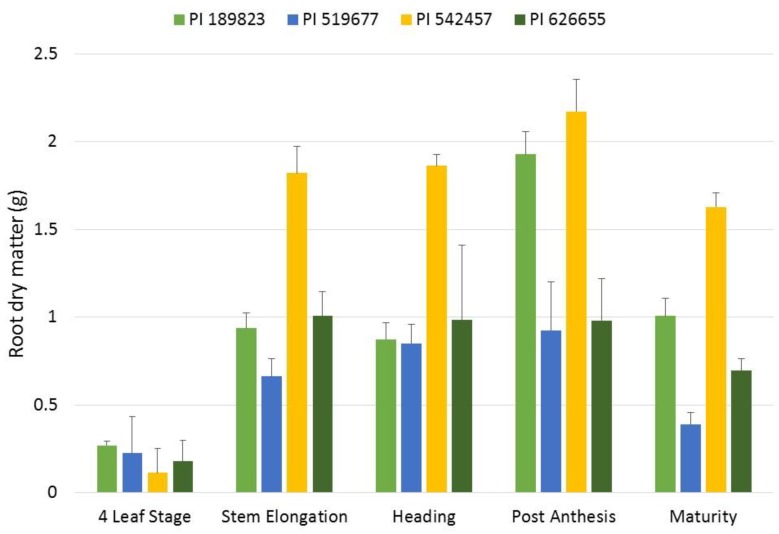
Increases in root dry matter in the four lines across the sampling points. Mean root dry matter for each of the four lines at the five growth stages sampled in the temporal assay are presented. Error bars represent the standard errors of the means. The different colors represent the different lines in the study shown in the legend above the figure.

**Figure 7 plants-09-00144-f007:**
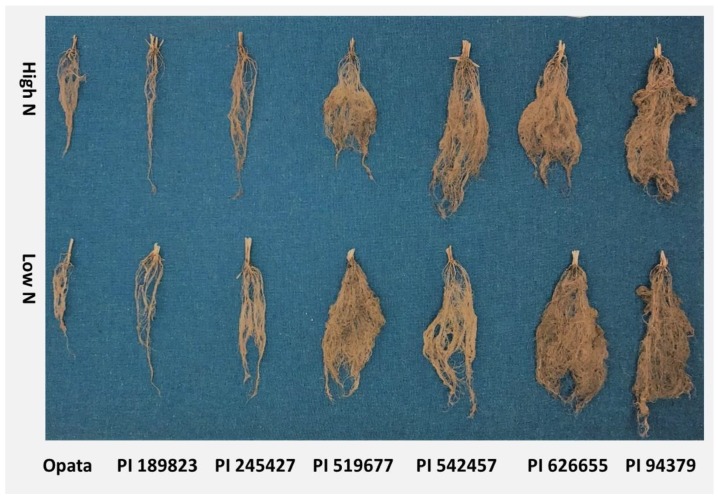
Representative root structure for each line in each N treatment from the maturity assay.

**Table 1 plants-09-00144-t001:** Differential responses of lines for eight traits in high N and low N environments in the maturity assay. The average, range, and repeatability of each trait was presented for each line. The significance of line, N treatment, and the line × N treatment interaction (L × T).

	Days to Heading (#)		Plant Height (cm)		Shoot Dry Matter (g)		Root-Shoot (g/g)		Root Dry Matter(g)		Yield (g)		Kernel Number (#)		Grain N Concentration (g N/g DM)	
	High N	Low N	High N	Low N	High N	Low N	High N	Low N	High N	Low N	High N	Low N	High N	Low N	High N	Low N
Opata	39	40	57.9	63.2	1.74	1.80	0.16	0.09	0.27	0.15	1.82	2.26	43	59	56.7	57.6
PI 189823	38	38	101.4	97.5	2.42	2.39	0.08	0.04	0.20	0.09	1.94	2.13	53	58	69.1	59.9
PI 245427	44	51	77.0	94.9	1.67	2.57	0.26	0.32	0.47	0.91	1.85	2.42	37	55	51.4	48.9
PI 519677	35	36	47.9	53.9	0.98	1.03	0.12	0.11	0.12	0.12	1.73	1.92	56	62	47.2	48.0
PI 542457	62	64	100.8	94.6	5.08	4.10	0.32	0.34	1.66	1.39	2.28	1.86	107	93	61.1	39.1
PI 626655	57	60	102.9	100.7	5.50	4.34	0.33	0.35	1.81	1.56	3.17	2.69	141	112	70.2	57.5
PI 94379	53	57	105.4	104.6	4.25	3.36	0.25	0.18	1.06	0.62	2.55	2.15	67	55	63.4	49.1
Average	46	49	84.5	86.8	3.07	2.77	0.21	0.2	0.78	0.68	2.2	2.2	72.06	69.78	59.9	51.4
Range	35–62	36–64	47.9–105.4	53.9–104.6	0.98–5.49	1.03–4.34	0.08–0.33	0.03–0.35	0.12–1.81	0.09–1.56	1.73–3.17	1.86–2.69	37–141	55–112	47.2–70.2	39.1–59.9
Repeatability	86.79%	77.18%	83.08%	85.13%	85.76%	73.87%	50.20%	62.86%	69.74%	72.41%	25.24%	3.01%	73.87%	49.35%	87.13%	95.45%
Line	***		***		***		***		***		ns		***		***	
N Treatment	ns		ns		ns		.		ns		ns		ns		*	
L * T	ns		*		**		ns		ns		ns		ns		**	

*p* value significance of as follows: *** < 0.001; 0.001 < ** < 0.01; 0.01 < * < 0.05; 0.05 < . < 0.1; ns > 0.1.

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
