# Peer review of "Variation in Root and Shoot Growth in Response to Reduced Nitrogen"

_plants, 2020, doi:10.3390/plants9020144_

Round 1

Reviewer 1 Report

The manuscript entitled “Association of Root Growth with Nitrogen Content and Individual Plant Gran Production in Wheat” presents results about how roots traits vary between 7 wheat lines in high and low N environment. While the experimental design is sound, there are major issues with the manuscript:

The aim of the study is not properly justified. As there are already some studies who focused on wheat lines and how their root and shoot traits vary (you notably cite Ehdaie et al. 2014), It is not easy to catch the novelty of the work presented here. Moreover, it is said that authors focus on root dry matter content (l.65), but this is not justified why it is interesting and/or particularly unstudied? Overall, it is as well difficult to identify the novelty of the article in the results and discussion section. The overall structure of the article is not easy to follow. In particular, the statistical analyses would be clearer if presented in a separate paragraph in the material and methods. I think as well that merging results and discussion makes the article not easy to read. The discussion is rather weak and most often cite article but do not make a clear link to the present results or just mention that the present results confirm previous studies. This is not sufficient, you have to show what novel understanding it brings. The general conclusion for breeding program is too vague. How your study does help to identify which line is more adapted to high N and low N environments? Do not simply state that such studies will help to develop breeding programs, this is too vague. In general, avoid using so many abbreviations (SDM, SNC, RDM, SN, GNC, etc). It is very hard to follow as nearly all your traits are named by abbreviations, even in the text. Replace by consistent explicit names.

Detailed comments:

Introduction:

l.58: Which root/shoot ratio (high or low) is beneficial in low N environment?

l.61: I am personally not familiar with the term Germplasm. Maybe you could say “different genetic lines”, which is easier to understand to a wider audience.

l.62: It is not clear to what “these phenotypes” refer to? Explicit that your test different genetic lines of Wheat and that you expect that these different genetic lines will express contrasting phenotypes. Notice that it is important that you mention that you focus on Wheat in the aim of the study.

l.68: I suggest you change experiment by “part” (here and throughout the manuscript).

Material and methods:

l.79: Please use the “Argentina line” and not only “Argentina” as it is as your are talking about the country of Argentina… and not the wheat line from Argentina.

l.103: Why roots are preserved in ethanol and not shoot? If you want to measure biomass, I think you don’t need to preserve roots in ethanol.

l.113-120: The statistic explanation is too messy. I understand you have pseudo replications. What you could do is to include “pot” as a random factor. Please say what you did and not what was not possible to do (ANOVA, according to you). R studio is an interface to work with R, please cite R (a statistical software) and not R studio. If you group all the statistical analyses in a paragraph, you can end this paragraph by indicating which software you use, this would avoid repetition. This is as well not an information of high priority, so it is nice it is mentioned at the end. The number of replicate and the overall design of the experiment must be given up front and not at the end of the description of the experiment as you are doing here.

l.125: “Would” is not appropriate to describe an objective

l.127: remove one “in”

l.140: Replace “collected” by “measured”

l.146: “shoot” and not “shot”

l.148-158: There part is to wordy. Be more accurate and succinct. You did an ANOVA, followed by Tukey post-hoc test.

l.159: Replace “dynamic” by “temporal” here and throughout the manuscript.

Results:

l.187-188: This first line is an aim not a result. Please delete.

l.188-191: It looks like you are providing the plan of your results-discussion part but afterwards it was not following this plan, so I found these first sentences confusing.

l.187-195: This first paragraph does not fit with the title of this part (Greater variation in belowground traits compared to aboveground traits.

Figure 1: Please provide a clear title and not a long explanation of why you didn’t do statistical tests here. Moreover, even with pseudo replicates, you should be able to test statistically the difference.

l.203: I am not sure that genetic variation is appropriate here. You observed phenotypic variability, which is due to genetic diversity. Please make sure we understand it this way. Genetic variation may indicate the genetic variability (variability of the genes) and not of their associate phenotypes.

l.211: “At seedling stage” and not only “at seedling”

Table 1: Be more succinct in the legend of your table. Moreover, you should provide F value and P value for each Anova and make sure we understand which treatments were compared. The idea to provide average value between high and low nitrogen is not convincing. Let the reader to see the original data. Indicate only the relevant significance levels (for example, I can’t see any **).

228: why it is important to provide so much details about the material and methods of this study?

l.238-240: This sentence does not make the link between the cited study and the present study.

243: RDM explained 78% of the variation in RTS: This is an interesting result!

l.32: Replace “association” by “relationship”

l.257-260: In what this cited article helps to understand your present results?

l.261: Do not refer to statistic here, this should be done in Material and methods.

l.21-l.270: This is too wordy and not clear! It seems to me that the last sentence (l.270-271) is a good summary of what you aim to say here. But then, the following paragraph seems as well to be kind of a repetition of this paragraph. Please be more straightforward and highlight only the essential results.

l.276-281: The link between this cited article and your study is clearer but if your results are just in line with previous results, it is hard to see what novelty is brought by your work. The same remarks holds for l.292-297.

l.301, remove ().

l.307: “the lines from Chile” and not “Chile”

Figure 5: It is always possible to draw a line between two groups of points… it does not mean there is a linear relationship between the data… be careful.

Figure 6: You should provide ANOVA F and P values for each test you did.

l.326: Starting with “in this study” is confusing as you are presenting your results here so it should be obvious.

Author Response

We would like to thank this reviewer for their time and great comments that were made. The text was changed in a major way thanks to these comments. We have removed abbreviations for traits as we see that it could be difficult to follow. Additionally, we have split the results and discussion sections in agreement with this review. While we still did not feel comfortable performing ANOVA using the pseudo-reps (this was discussed in Seth’s thesis committee in detail and we want to honor those discussions as well), we did feel comfortable performing a permutation test to differentiate the N treatments. Much of the text has been shifted making it difficult to respond to specific comments on particular lines; however, we have evaluated the text and did our best to make all the needed corrections in agreement with this review.

Reviewer 2 Report

The authors analyzed the relationship between root dry matter and yield or nitrogen uptake in environments of high or low nitrogen.  The authors aimed to characterize nitrogen uptake in seven different wheat varieties as a function of the environment and root dry matter.  Experiments are conducted in a greenhouse setting.  The authors identify that in high nitrogen conditions, a larger root dry matter translates into increased nitrogen uptake and higher grain yield.  Overall, this manuscript provides some interesting findings that need to be validated in a field setting.   

Major points:

Line 290-291: Use of the word “efficient” is subjective based on the data presented here.  The data shows that the yields are independent of RDM under low N and total N is not significantly different.  Therefore, one could conclude that when under low N conditions, there is not a negative tradeoff to the allocation of more resources to the roots.  In fact, this indicates that the larger RDM would be preferred because it far outperformed during high N.  This is an important note, because it is opposite of previous reports.  The authors should justify why they came to the conclusion they have stated and discuss why they didn’t see a negative consequence of the larger RDM as previously reported.  Figure 5: This data could be interpreted as 2 groups, instead of one group with a negative trend.  If the seedling RDM is below 12.5 mg, then the mature RDM is much greater than if the seedling RDM is greater than 12.5 mg.  This interpretation also agrees with Figure 4 during high nitrogen.  The authors should consider altering the emphasis they put on a negative correlation or provide justification as to why these should be treated as 1 group instead of 2 groups.  If in fact there are 2 groups, then a smaller seedling RDM is preferred and there needs to be more discussion about how one would want a seedling with smaller RDM for increased yields.  The 2-group interpretation is also opposite of lines 21-22 in the abstract.

Minor points:

Figure 1 needs a title. The first sentence is not needed. Please add significance to table 1 seedling stage for RDM and RTS. Please put the SDM data in table 1 or somewhere because this allows one to compare the relationship between the overall size of the plant and the RDM. Please include both high and low nitrogen RDM data for maturity in table 1. Plant height, tiller number, kernel number and spike dry weight are discussed in the methods, but results are never shown or discussed. Was this intentionally left out? Line 226-227: statement about above-ground organs is not supported by Figure 2. Are there any pictures of the shoots or additional data that is missing?  If not then, consider changing this wording. Line 239-240: RDM at the seedling stage has a 2-fold difference between the extremes, suggesting that root differences are already present.  Please clarify the sentence.  Line 272-276 are essentially repeated from Lines 261-271. Figure 5: Indicate which genotypes are used for the dynamic experiment. The legend also states colors, but no colors are used.  Figure 6: With the exception of USA, why are the maturity numbers so different from Experiment 2? Also include in the legend that this is high nitrogen. 

Author Response

We would like to thank this reviewer for their thorough comments to improve our manuscript. We felt like an area that was lacking in the eyes of the reviewer was our portrayal of the difference of the phenotypes due to the low N treatment. As such, we have worked to strengthen this area of the manuscript. Furthermore, while we felt that separating the lines into large-, medium-, and small-rooted lines made for the best story, we respect the opinion of the review and we have removed that part of the story. Instead we focused on the trait correlations between the 2 N treatments.

Reviewer 3 Reports

I am delighted to see phenotypic studies of this sort being carried out. I really thought that the research rather missed the mark by firstly designing expt 1 in such a way that the lines were not truly replicated and therefore between line comparisons could not be made. I also did not understand why only high N was used in the more detailed study. We are desperately needing information about how different germplasms perform under low nutrient conditions. here you had a chance to elucidate these lines and really we need to see the performance under poor nutrient conditions. However - the research has been completed and therefore the most has to be made of it and in this case I did not feel that the most was made of the results obtained. We really need to hear about the response to low N - I made a table from the only data set given in the paper looking at the difference between High N and low N which showed three lines (Mexico, USA, Iran) all either zero or +only 1mg RDM; two which had a negative effect (Armenia and Afghanistan) with -2mg and two with + effect >3mg (Argentina and Chile). Nowhere did we see the discussion of the RESPONSE to N limitation (although you did discuss other research where it was considered). It was difficult to review as one set of data was missing (table 1) and by amalgamating the data into large, medium and small all the information relating to the individual lines and how they behave was sadly lost.

Author Response

We greatly appreciate the insights from this reviewer. We have updated the manuscript to include the major comments that the reviewer has made. We especially found your point interesting about large root dry matter not being a drag in low-N treatments, but it was good in high-N treatments. This has also been added in the discussion section of our manuscript. In reference to the second major point that this reviewer has mentioned, we have decided to shift our focus away from looking at the regression of root dry matter between the seedling stage and maturity stage studies. We did this because regression was only based on the 14 means for the lines in both treatments. We felt that there might have been the potential for a better study elsewhere in our manuscript.

Round 2

Reviewer 3 Report

This is greatly improved and I like one of the new figures (figure 3) - very helpful and clear.  However figure 4 is not useful in my view, figure 5 is very helpful to look at the mean responses and the matrix in figure 6 is useful but possibly over complex (difficult to read).  However, in my opinion, this paper could still be improved.  The main point I made in my first review has not really been addressed.  Namely

"Nowhere did we see the discussion of the RESPONSE to N limitation (although you did discuss other research where it was considered).

I think the results could be looked at again in the light of the main questions -

how much variation in root traits is there amongst these diverse 7 lines of wheat?

how do the various lines respond to N depletion and is there any relationship with the response and the difference in yield between high and low N (not discussed at all and nowhere were the various yields given (sadly)).”  (review 1)

The key here is to see that the lines differ in their response to N availability and rather than present the mean effects (although the ranges are always helpfully given) identify candidate lines for further investigation. 

 If figure 4 is removed there would be room for a figure or table presenting the various lines and their yields in relation to N response.  The emphasis has been improved, and there is now a really useful screen suggestion at growth stage 2, but is still not picking out the differential responses of each line and in fact the number of figures showing the lines has been reduced overall with the removal of table 1 and the regression (figure3 version 1).

Author Response

Dear Editor & Reviewer

Responses to the comments:

We greatly appreciate the feedback from the reviewer. In response to the suggestions we Removed figure 4 (permutation test results) To accommodate this area that we added to, was removed which was also suggested. We emphasize the various responses of the lines to the N limitation. Now, there is a new figure (6) that emphasizes the changes in root and shoot dry matter in addition to the change in yield between the N treatments. We agree that this is an area that there needs to be more literature. We have also made figure 5 (correlation matrix) easier to understand. We included less traits in this matrix and we made all of the text bigger. Additionally, we added a note that the correlations shown above the diagonal are from the low-N treatment and those below the diagonal are from the high-N treatment.

Comments:

This is greatly improved and I like one of the new figures (figure 3) - very helpful and clear.  However figure 4 is not useful in my view, figure 5 is very helpful to look at the mean responses and the matrix in figure 6 is useful but possibly over complex (difficult to read).  However, in my opinion, this paper could still be improved.  The main point I made in my first review has not really been addressed.  Namely

"Nowhere did we see the discussion of the RESPONSE to N limitation (although you did discuss other research where it was considered).

I think the results could be looked at again in the light of the main questions -

how much variation in root traits is there amongst these diverse 7 lines of wheat?

how do the various lines respond to N depletion and is there any relationship with the response and the difference in yield between high and low N (not discussed at all and nowhere were the various yields given (sadly)).”  (review 1)

The key here is to see that the lines differ in their response to N availability and rather than present the mean effects (although the ranges are always helpfully given) identify candidate lines for further investigation.

 If figure 4 is removed there would be room for a figure or table presenting the various lines and their yields in relation to N response.  The emphasis has been improved, and there is now a really useful screen suggestion at growth stage 2, but is still not picking out the differential responses of each line and in fact the number of figures showing the lines has been reduced overall with the removal of table 1 and the regression (figure3 version 1)

Round 3

Reviewer 3 Report

This paper has been improved greatly with the addition of new figures. However there are still details which need attention. I also feel that the original table of data which was in version 1 (including the results from the individual lines) is more informative than presenting means and ranges. The important results are now more clear - but could still be clearer. Which lines have the best potential for a future in low N environments? It is clear you achieved interesting results as there are very different strategies displayed by the lines (the new figure shows this nicely). However as we still don't have that information related to the overall success of the lines (which produced the greatest yields? were the best lines the same at each level of N? This is what we are left wanting to know.
It is nearly there now.

Author Response

Response to Reviewer

1- I also feel that the original table of data which was in version 1 (including the results from the individual lines) is more informative than presenting means and ranges.

Thank you for your kind words and your help in making it what it has become. All changes that were made in this iteration have been highlighted to distinguished them from previous changes. We added the means into the new table and combined the information. We were concerned with burdening the reading with too much information, however, we agree that this additional information does help tell the story of our work.

2- Which lines have the best potential for a future in low N environments?

To address this question, we added a paragraph in the results and a paragraph to the discussion sections. We also added a sentence to the conclusion discussing the top performing lines for each N treatment.

3- We still don't have that information related to the overall success of the lines. Which line produced the greatest yields? Were the best lines the same at each level of N?

This is now presented in the new table 1. I also think that I discussed this more in-depth in this iteration.

Additional work

There was an additional comment that the reviewer preferred the line identification based on the geographical locations of the lines. At this time, we would not like to make this change back. Another review that we received for this manuscript made a good point in saying that these lines do not necessarily represent all lines from these locations. They were confused as they thought that was what we were trying to say. In other words, they told us that they found that distracting and took the focus away from the work.

In the graph that this is commented in, Figure 6, we chose to use numbers 1-7 to represent the 7 lines. This was done because when we used a longer identifier (such as PI number) the names ran into each other making it difficult to read.

There was also a mistake that was found in the text. Originally, we had the correlation between root dry matter and grain N concentration in low in as positive whereas now the correlation is negative. We regret that this mistake was made, but we are happy that it was corrected before further evaluation.

Round 4

Reviewer 3 Report

This is much improved and there are just a few points I would like you to check over which are highlighted with a few comments.

I am hopeful this work leads to a larger field trial testing of some of your extreme lines which might really help to improve the wheat lines of the future as we face changing agriculture to address climate change.

Author Response

Response to Review

We are very grateful for the detailed comments of the reviewer to strength this work. We have made many of the adjustments that you have requested.

I don’t think this statement is needed.

The statement on line 154 – 155 “Since there were only two N treatments no further post-hoc tests were performed to separate N treatments when the difference was significant.”  has been removed in response to the reviewer.

Not necessary to explain as the values on the axes are all positive. The figure is very clear.

The statement on lines 203-204 “However, we still presented the amount of dry weight below horizontal axis as positive values.” was removed in response to the reviewer.

Very informative to see each line and the overall significances. What is the significance of Root-Shoot for N treatment? Is that a “close to significant” sign? Is seems odd that so few of the measures show significant effect on N treatment?

In response to the first comment, we added a line after the table describing the significance codes for the ANOVA. You also comment on the minimal amount of significance that we had for N treatment. Later you made a comment that the large genetic variability might be a player in the smaller effect of N, and we agree with this comment. As such, later we have added a sentence to describe this as a possible partial cause.

But not significantly for any of the lines? It might be helpful to identify if there was no significance due to high variability in the measurements? No sds are presented (they would make the table too cluttered and you give the significances) but maybe just a comment about why so few are significant given the seemingly large effect of the N treatment on the growth patterns.

I added in the sds for the shoot dry matter measurements for each N treatment. I added 2 reasons for the lack of significance given the seemingly large impact on the growth pattern. The first being the differences between the responses of the individual lines and the second being the standard deviation within the N treatments.

There is no ns for root-shoot ratio in the table – I thought it was showing marginally significant effect of N treatment? Make this clear in the table.

While it is true that the p-value was between 0.05 and 0.1, I still thought that this should be considered not significant. I do add the p-value though for which we declared it not significant.

This sentence does not make sense. Please check it. Yield was based on …

I made a minor change to this sentence to improve the clarity. I changed and in the center of the sentence to rather than. This shows that we were comparing the analysis done here to those in the field with plant-to-plant interaction.

Although the lines were highly significantly different – no significant effects of the N treatment except in grain N conc. This is a bit surprising but possibly needs a comment here? Could it simply be that the lines you chose to examine are very genetically variable?

We added in a sentence (line 342-344) in the paragraph described to state that the genetic variability was one of the reasons for the limited significance of the N treatment. We think that this statement helps to improve the clarity of the work.

Where it is likely that nitrogen management practices will be adjusted to prevent leaching and environmental damage.

We added this sentence to the end of the manuscript in agreement with the comments of the reviewer.

Editor Response

Thank you for taking the time to review this work. We are grateful for your comments and hope that our response is sufficient for you to understand our reasoning behind our decision.

In the maturity-test, the pots were filled with a potting mixture including one-third topsoil from an agricultural source in Indiana. This component might have contained enough N to mask the difference between the two treatments. Would the authors please provide data of N content in the soil mixture to ascertain the difference between the Low-N and the High-N environments. This will help the readers assess the conclusions they have reached.

Thank you very much for taking the time to evaluate our work. In this study we ended up not doing an analysis on the soil. We did consider this approach however we were not so much interested in the nitrogen content in the soil, but we were more interested in the amount of nitrogen that was in the grains at maturity. We felt like this was a more accurate representation of the presence of the nitrogen treatments throughout our study. As such, we do not have the information on the N content in the soil mixture.
